# PHYSICS-INFORMED MACHINE LEARNING FOR FLUID FLOW PREDICTION IN POROUS MEDIA

**Ali Takbiri-Borujeni**
Amazon
Seattle, WA
alitakb@amazon.com

**Mohammad Kazemi**
Department of Engineering
Slippery Rock University
Slippery Rock, PA, 16057
mo.kazemi@sru.edu

**Sam Takbiri**
University of Isfahan
Isfahan, Iran
sam.takbiri@gmail.com

## ABSTRACT

The objective of this work is to predict grid-level flow fields in porous media as a priori to determining the permeability of porous media. A physics-informed ML model is developed by using the results from numerical fluid flow simulations of randomly distributed circular grains to represent the porous media. The deep U-Net and ResNet neural network architectures are combined to train a deep learning model that avoids vanishing gradient issues. The model integrates continuity and momentum conservation equations into the loss function to ensure physical consistency. Additionally, we modify the padding function in convolutional layers to use circular paddings, mimicking periodic boundary conditions in LB simulations. By learning inter-grid communications, the ML model achieves precise flow predictions for new simulation sets with high accuracy. The robustness of the developed model is then tested for numerous variations of porous media that have not been used for developing the model.

## 1 INTRODUCTION

Rock permeability plays a crucial role in various fields of engineering science. Accurate estimation of rock permeability is essential for reservoir engineering (especially in the petroleum industry), geotechnical engineering, hydrogeology, and environmental projects Hughes (2013); Kazemi & Takbiri-Borujeni (2015); **?**. Understanding rock permeability helps optimize fluid flow, predict groundwater movement, and design effective reservoir management strategies. Traditional methods for determining rock permeability and flow properties struggle with complex rock morphology and heterogeneity. High-fidelity numerical simulations based on image data, including the lattice Boltzmann Method, offer promise but remain computationally intensive (Sukop, 2006; Borujeni et al., 2013). Leveraging recent advances in ML presents an opportunity to reduce computational costs while preserving physical interactions (Mo et al., 2019; Zhu & Zabaras, 2018).

With recent advances in data-driven methods, promising new approaches, have been introduced, significantly reducing the computational cost of the numerical simulations while preserving the fluid-fluid and fluid-solid physical interactions. Mo et al. (2019) developed a deep encoder-decoder convolutional neural network (CNN) for uncertainty quantifications of dynamic multiphase flow. Zhu & Zabaras (2018) used the Bayesian neural network to predict the pressure and velocity fields in a channelized permeability field. Araya-Polo et al. (2019) and Sudakov et al. (2019) developed a deep learning model to predict the permeability from images of porous structures. Sun et al. (2020) proposed a physics-constrained deep learning approach that enforces the initial and boundary conditions and the Navier–Stokes equations in the loss function to predict the flow velocities in various geometries. Several other implementations of deep learning have been published in recent years Kamrava et al. (2019); Mosser et al. (2017); Tian et al. (2020).

Building data-driven models requires computationally expensive lattice Boltzmann or numerical Navier-Stokes simulations. While post-processing these simulations can yield permeability values, a significant amount of high-fidelity data (e.g., velocity vectors at grid blocks) is lost. (Takbiri-Borujeni et al., 2020) introduced a deep learning algorithm that predicts pixel-scale velocity vectors for randomly generated 2D circle packs. This data-driven surrogate model replaces the numerical

simulator, enabling accurate predictions without new lattice Boltzmann simulations. The model, based on pixel-level velocity vectors computed using LBM, employs a CNN with contracting paths and residual blocks. Despite achieving less than 15% error, the velocity profiles remain non-smooth.

This work builds upon the work done by (Takbiri-Borujeni et al., 2020) by developing a physics-informed ML model to predict pixel-level velocity fields in randomly generated circle packs. By embedding known conservation laws into the loss function, this approach significantly improves the smoothness and accuracy of grid-level velocity profiles compared to previous methods Takbiri-Borujeni et al. (2020).

## 2 METHODS

### 2.1 LATTICE BOLTZMANN SIMULATIONS

This research employs lattice Boltzmann (LB) simulations, known for effectiveness in irregular geometries and parallelizability (Sukop, 2006). LB models fluid behavior with discrete lattice meshes, governed by the Boltzmann equation. For more detail on LB simulation refer to the Appendix.

## 3 DEEP LEARNING MODEL DEVELOPMENT

To incorporate physics into the proposed ML model, changes are made to the architecture and loss function of our former model (Takbiri et al., 2022; Takbiri-Borujeni et al., 2020). This includes switching padding from reflective to circular to mimic LB simulations' periodic boundary conditions and adding divergence-free requirements to the loss function. To address non-smooth velocity predictions, circular padding was implemented to mimic periodic boundary conditions, ensuring values on opposite sides of the domain were close. Figure 1 compares velocity predictions in the x-direction using reflective padding from our previous work (Takbiri-Borujeni et al., 2020) with modified circular padding. Circular padding improves accuracy, particularly near boundaries.

### 3.1 INCLUDING PHYSICS IN ML

Physics-informed machine learning combines deep learning algorithms with physics, employing various integration techniques. Data for model training can originate from physical systems via experiments or simulations. Integration levels range from incorporating physics into loss functions to interleaving full physical simulations with machine learning outputs, particularly beneficial for systems with temporal evolution. This work builds upon previous research (Takbiri-Borujeni et al., 2020) to address limitations observed in velocity profiles. By integrating physics into the model architecture and modifying the loss function to reduce grid-level fluctuations, this work improves velocity predictions.

#### MODIFYING THE LOSS FUNCTION

Consider that the numerical domain constructed by spatial locations $\mathbf{s}$, where the pixels of the binary input image $x(\mathbf{s})$ and $\mathcal{S} = \{s_1, ..., s_{n_s}\}$ are the index set for the spatial grid locations, $s \in S \subset R^{d_s}(d_s = 1, 2, 3)$ are the spatial locations. The simulations can be considered as mapping of $x \in \{0, 1\} \subset R^{d_x n_s}$ to its corresponding solution $\mathbf{u} \in \mathcal{U} \subset R^{d_\mathbf{u} n_s}$,

$$\eta : 0, 1 \rightarrow \mathcal{U} \,, \tag{1}$$

where $\mathbf{u} = \eta(x)$. The purpose for building the surrogate model is to develop a new mapping function, $\hat{\mathbf{u}} = \mathcal{F}(x, \boldsymbol{\theta})$, to be trained using a limited number of simulation data, $\mathcal{D} = \{x_i, u_i\}_{i=1}^N$ with $\boldsymbol{\theta}$ as model parameters and $N$ as the number of samples in the training, to approximate the predictions made by $\eta$ mapping.

The approach used in Takbiri-Borujeni et al. (2020) was to develop a model that estimates the target variable $\hat{\mathbf{u}}(x(\mathbf{s}), \boldsymbol{\theta})$. The loss function of the standard approach is thus minimized such that,

$$\mathcal{L}_{data}(\boldsymbol{\theta}) = \frac{1}{n} \sum_{i=1}^n ||\hat{\mathbf{u}}_i(\boldsymbol{\theta}) - \mathbf{u}_i||_1 + \lambda \Omega(\boldsymbol{\theta}) \,, \tag{2}$$

$$\boldsymbol{\theta}^* = \arg\min_{\boldsymbol{\theta}} \ \mathcal{L}_{data}(\boldsymbol{\theta}), \tag{3}$$

where $\Omega(\boldsymbol{\theta})$ represents regularization (a measure of complexity of the model), and $\lambda$ is the regularization strength, $\boldsymbol{\theta}^*$ denotes a set of (sub)optimal weights obtained from the optimization

Here, we consider leveraging the governing equations in the loss function so that governing equations for the model predictions, $\hat{\mathbf{u}}$, are enforced (Sun et al., 2020). The loss function, $\mathcal{L}_{phy}$, is defined by enforcing the governing equations Karpatne et al. (2017).

To develop $\mathcal{L}_{phy}(\boldsymbol{\theta})$, the mass and momentum conservation equations are considered. It should be noted that Navier-Stokes equations can be derived from LB models using the appropriate local equilibrium distribution function (Chen & Doolen, 1998). The mass conservation equation for incompressible fluid turns into a divergence-free condition. However, since the density is an intrinsic variable in the LB equation, all LB simulation schemes are compressible to some extent and are never divergence-free (Luo et al., 2011).

$$\nabla.\mathbf{u} = \mathbf{0}\,. \tag{4}$$

On the other hand, the flow of incompressible fluids is described through Navier-Stokes conservation of momentum equation,

$$\frac{\partial \mathbf{u}}{\partial t} + (\mathbf{u}.\nabla)\mathbf{u} - \nu\nabla^2\mathbf{u} + \mathbf{b}_f = 0 \tag{5}$$

where $\nu$ is kinematic viscosity, and $\mathbf{b}_f$ is the body force. Due to the steady-state condition of the flow, the term with time in the above equation is zero. Furthermore, for an incompressible fluid at a low Reynolds number, the equation reduces to,

$$\nabla^2\mathbf{u} + \mathbf{b}_f = 0\,. \tag{6}$$

The $\mathcal{L}_{phy}(\mathbf{W}, \mathbf{b})$ for the deep learning architecture is consisted of the continuity equations, Equation ( 4). Therefore, our loss function becomes,

$$\mathcal{L}_{phy}(\boldsymbol{\theta}) = \frac{1}{n}\sum_{i=1}^{n} \underbrace{||\nabla.\hat{\mathbf{u}}(\boldsymbol{\theta})||}_{\text{Mass conservation}} + \underbrace{||\nabla^2\hat{\mathbf{u}}(\boldsymbol{\theta}) + \mathbf{b}_f||}_{\text{Momentum conservation}}\,. \tag{7}$$

Figure 2 demonstrates the results for the same circular pack discussed in Fig. 1. The smoothness issue seems to be resolved significantly in the physics-informed case. There are some irregularities still observed in the profile.

Another loss function definition approach was implemented to improve the model further. In this approach, the mass and momentum conservation constraints for LB simulation results are subtracted from those for the model predictions, and thus, the modified loss function, $\mathcal{L}_{dev}$, is defined as,

$$\mathcal{L}_{dev}(\boldsymbol{\theta}) = \frac{1}{n}\sum_{i=1}^{n} \underbrace{||\nabla.\hat{\mathbf{u}}(\boldsymbol{\theta}) - \nabla.\mathbf{u}||}_{\substack{\text{Deviations from} \\ \text{Mass Conservation} \\ \text{constraints}}} + \underbrace{||\nabla^2\hat{\mathbf{u}}(\boldsymbol{\theta}) - \nabla^2\mathbf{u}||}_{\substack{\text{Deviations from} \\ \text{Momentum Conservation} \\ \text{constraints}}}\,. \tag{8}$$

The final loss function can then be written as

$$\mathcal{L}(\boldsymbol{\theta}) = \mathcal{L}_{data}(\boldsymbol{\theta}) + \mathcal{L}_{phy}(\boldsymbol{\theta}) + \mathcal{L}_{dev}(\boldsymbol{\theta})\,, \tag{9}$$

$$\boldsymbol{\theta}^* = \arg\min_{\boldsymbol{\theta}} \ \mathcal{L}(\boldsymbol{\theta})\,, \tag{10}$$

where $\boldsymbol{\theta}^*$ denotes a set of (sub)optimal weights obtained from the optimization. The next section describes the results of the final model.

## 4 OVERVIEW OF MODEL ARCHITECTURE

512 2D circle pack images of size $128 \times 128$ pixels are generated by randomly placing circles of radius 10 to 30 pixels within a 128x128 pixel grid domain. The number of circles varies from 1 to 10, resulting in porosity values ranging from 20 to 60%. LB simulations are then used to determine the velocity fields and permeability of these images.

The neural network architecture, referred to as U-ResNet, is inspired by U-Net and residual networks (Ronneberger et al., 2015; Hinton & Salakhutdinov, 2006). It consists of downsampling followed by residual blocks and upsampling layers (Hinton & Salakhutdinov, 2006). Convolutional layers, instead of pooling layers, are utilized for downsampling. Residual blocks enhance computational capacity at the end of downsampling.

The network employs five strided convolutions to reduce input size by 64, compressing multiple 3x3 convolutions into one 7x7 convolution, thus reducing computational costs. Multi-resolution features are used to capture long-range dynamics, downsampling the first hidden layer five times before upsampling. Circular padding is implemented for all convolution layers to model periodic boundary conditions in LB simulations.

Gradient vanishing with increasing network depth is mitigated by employing Resnet blocks (Zagoruyko & Komodakis, 2016). Evaluation metrics like RMSE and $R^2$ assess model performance, with Fig. 3 illustrating the network architecture.

Several metrics are considered to quantitatively evaluate models on validation data, namely, root mean squared error (RMSE), $\sqrt{\frac{1}{N} \sum_{i=1}^{N} \| \hat{\mathbf{u}}^i - \mathbf{u}^i \|_2^2}$, and the coefficient of determination ($R^2$), $1 - \frac{\sum_{i=1}^{N} \| \hat{\mathbf{u}}^i - \mathbf{u}^i \|_2^2}{\sum_{i=1}^{N} \| \bar{\mathbf{u}}^i - \mathbf{u}^i \|_2^2}$.

## 5 RESULTS

### 5.1 TRAINING, VALIDATION, AND TESTING

The ML model is trained on the input images and corresponding LB simulation results. All velocity values are normalized between zero and one. Model validation is conducted by evaluating its performance on the training data. Fig. 4 compares grid-level x-direction velocity components predicted by the physics-informed model with LB simulation results for two different input images. Contour maps of predicted velocity values closely resemble those from LB simulation. Fig. 4 shows the absolute error distribution at the grid level, indicating an accurate prediction of velocity values inside solid grains, near grain interfaces, and away from interfaces. Error distribution plots reveal negligible errors, with the majority within 2.5%. The physics-informed model is further tested by predicting velocity fields for images not included in the training set.

Grid-level results are presented for two test images, showing x-direction velocity in Fig. 5. Strong agreement is observed between the predicted velocity values by the physics-informed model and LB results, with grid-level errors within 6% and the majority within 2.5%.

Grid-by-grid comparison of velocity field components is discussed, revealing predictions by the physics-informed model closely matching LB simulation results, with velocity profiles demonstrating agreement within 10%. Additionally, the permeability tensor's diagonal elements closely align with the unit slope line across all test cases, indicating robust agreement between the physics-informed model and LB simulation results.

## 6 CONCLUSIONS

We develop a physics-informed ML model using numerical fluid flow simulations from randomly distributed circular grains to represent the porous media. The model combined deep U-Net and ResNet neural network architectures. The model integrates continuity and momentum conservation equations into the loss function for physical consistency. Additionally, we modify convolutional layer padding to use circular paddings, mimicking periodic boundary conditions in LB simulations. By learning inter-grid communications, the ML model achieves precise flow predictions for new

simulation sets with less than 5% error. Its robustness is tested across various porous media variations not used during model development.

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

## 7 APPENDIX

### 7.1 LATTICE BOLTZMANN SIMULATIONS

LB models fluid behavior with discrete lattice meshes, governed by the Boltzmann equation. The lattice Bhatnagar-Gross-Krook (LBGK) model simplifies this equation, incorporating collision terms with relaxation time parameters influencing fluid viscosity (Sukop, 2006). Different models like single-relaxation-time (SRT), two-relaxation-time (TRT), and multiple-relaxation-time (MRT) offer variations in collision approximation (Ginzburg, 2005; Lallemand & Luo, 2000).

The D2Q9 model represents a two-dimensional system with nine possible fluid movement directions. External forces, like body forces, affect fluid velocity in equilibrium distributions (Sukop, 2006; Takbiri Borujeni, 2013). LBM simulations apply body forces until kinetic equilibrium, extracting velocity fields for further model development (Sukop, 2006; Takbiri Borujeni, 2013). A body force of $10^{-7}$ in lattice units is used in x direction, with simulations running until kinetic equilibrium, and velocity fields are extracted for model development.

### 7.2 FIGURES

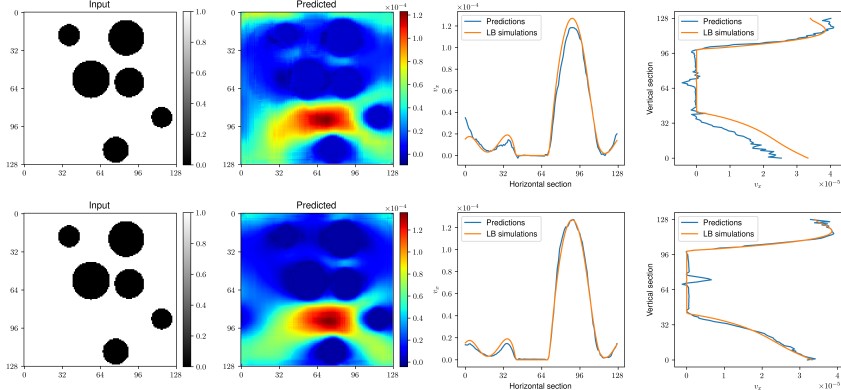

Figure 1: Comparison of models trained using reflective (top) and circular (bottom padding).

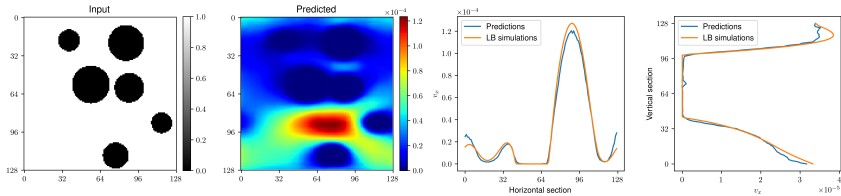

Figure 2: Comparison of models trained using reflective (top) and circular (bottom padding.

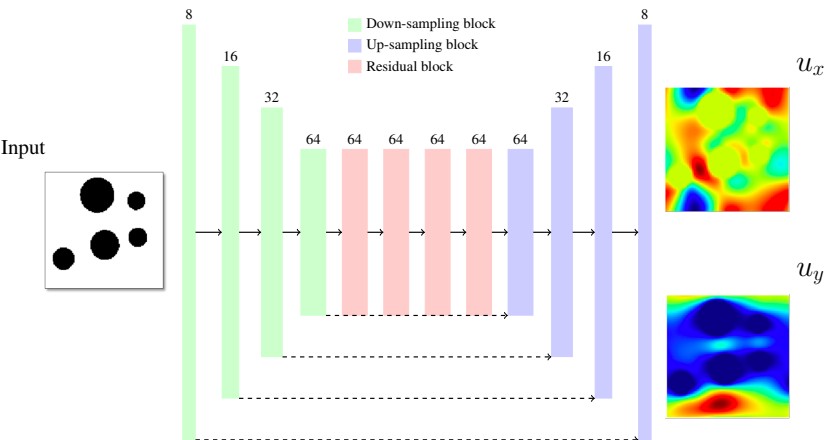

Figure 3: The architecture of the deep learning model used in this study.

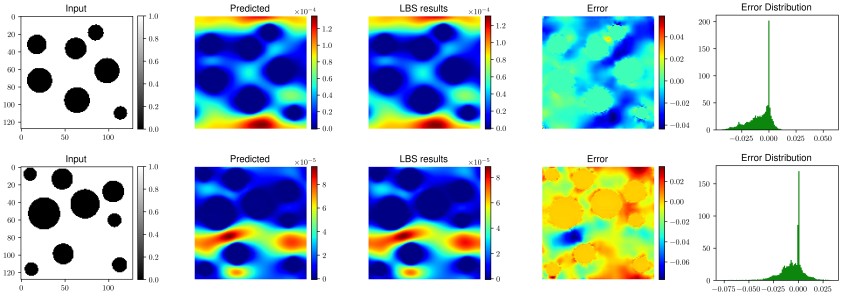

Figure 4: Training and Validation: Grid level comparison of the x-direction velocity with x-directed flow predicted by the ML model and the LB simulation result for the two training samples are shown. Both the grid level absolute error percentage between the ML model prediction and the LB simulation and the distribution of the absolute error percentage are shown. Velocities are in lattice units $[lu]$.

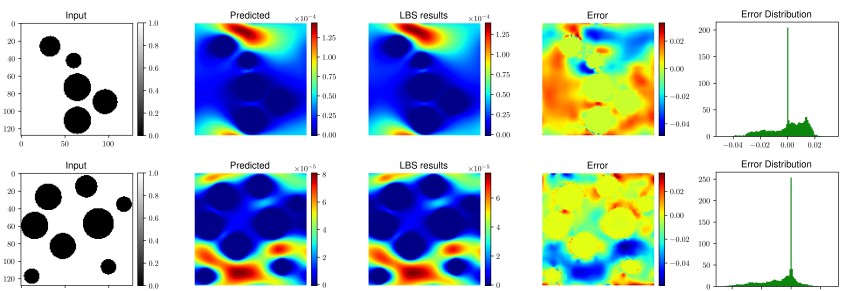

Figure 5: Testing: Grid level comparison of the x-direction velocity with x-directed flow predicted by the ML model and the LB simulation result for the two training samples are shown. Both the grid level absolute error percentage between the ML model prediction and the LB simulation and the distribution of the absolute error percentage are shown. Velocities are in lattice units $[lu]$.

