# OpenReview forum: "Physics-Informed Machine Learning for Fluid Flow Prediction in Porous Media"
_ICLR.cc/2024/Workshop/AI4DiffEqtnsInSci — AI4DiffEqtnsInSci @ ICLR 2024 Poster_

### Official Review · Reviewer_JDXF · 2024-02-16
**The topic is interesting; and novelty of the work is acceptable. However the results are not clearly demonstrating the novelties applied.**

**Rating:** 6
**Confidence:** 4

**Review:**

The paper showed the application of a physics-based deep learning approach in the pore-scale simulation of single-phase flow in porous media, as a replacement for the Lattice-Boltzmann (LBM) approach. It seems the improvements applied in the training algorithm could bring some advantages compared to the basis model (previous work). I do believe the applied changes are very useful in improving the results, however, the importance is not shown very clearly. Also,

	I understand the importance of developing reliable deep-learning methods for the simulation of pore-scale flow processes. I also understand the novelty of the work in using physics-based algorithms (PINNs). However, I believe the positive impact of using PINNs compared to the previous work (basis) is not shown properly. I think there could be cases that PINNs could bring much better responses compared to the basis model, to demonstrate the novelty of the work. It is important since adding a physics-based loss term adds significant time in the training of the model.
	As far as I understood, there were two changes in the model compared to the basis model: using circular padding, and physics-based loss function. The relative advantage of each is not clear by looking at the results. However, I understand the limitations in the extended abstract, maybe it can be applied to the full paper later.
	Equation 8 seems to be unclear to me. Did you use the residuals (PDE errors) from the LBM simulations? Why it needs to be applied if it is really an error?
	should you add parentheses in eqs. 7 and 8 to apply the impact of sigma two both terms?
	The paper starts with the importance of the measurement of permeability, … while in the results there is no information about the permeability predictions. It would be even better if there were a report about the improvements in the permeability prediction compared to the basis method.

	There are a variety of writings and referencing issues, such as 4th line in the introduction section.
	Unclear sentence: “With recent advances in data-driven methods, promising new approaches, have been introduced, significantly reducing the computational cost of the numerical simulations while preserving the fluid-fluid and fluid-solid physical interactions.”
	The authors mention the term ‘pixel-level’, while it is not really representative of the author's meaning, I guess. At least at some point in the abstract, I prefer to clarify it by a more understandable term such as ‘pore-level’ or ‘pore-scale’ …
	Figure 2 does not seem to show what has been mentioned in the caption.
	Figure 5: it is not clear if it is showing testing data or training data.
	The size of the dataset used for the training is not clear enough.

---

### Official Review · Reviewer_EK2Y · 2024-02-20
**Physics-Informed Machine Learning for Fluid Flow Prediction in Porous Media**

**Rating:** 4
**Confidence:** 2

**Review:**

This work presents a simulation of porous media using physics-informed ML models. The idea is not new and has already been done in the literature; see, for example,

1. Faroughi, S. A., Soltanmohammadi, R., Datta, P., Mahjour, S. K., & Faroughi, S. (2023). Physics-informed neural networks with periodic activation functions for solute transport in heterogeneous porous media. Mathematics, 12(1), 63.
2. Lehmann, François, Marwan Fahs, Ali Alhubail, and Hussein Hoteit. "A mixed pressure-velocity formulation to model flow in heterogeneous porous media with physics-informed neural networks." Advances in Water Resources 181 (2023): 104564.
3. Faroughi, Salah A., Ramin Soltanmohammadi, Pingki Datta, Seyed Kourosh Mahjour, and Shirko Faroughi. "Physics-informed neural networks with periodic activation functions for solute transport in heterogeneous porous media." Mathematics 12, no. 1 (2023): 63.

---

### Meta-Review · Area_Chair_pwAe · 2024-03-02

**Recommendation:** Accept (Poster)

**Metareview:**

The paper showed the application of a physics-based deep learning approach in the pore-scale simulation of single-phase flow in porous media, as a replacement for the Lattice-Boltzmann (LBM) approach. The work builds upon prior work in this area and whilst it is an interesting development, there is a lack of details on exactly how it is improving and the differences to prior methods. It is useful however to the community and should be accepted for a poster session.

---

### Decision · Program_Chairs · 2024-03-02

Accept (Poster)